# Geographic distribution of nematodes in the Atacama is associated with elevation, climate gradients and parthenogenesis

Laura Villegas [1] ✉, Laura C. Pettrich[1], Esteban Acevedo-Trejos [2,3], Arunee Suwanngam[4], Nadim Wassey[1], Miguel L. Allende [5], Alexandra Stoll[6], Oleksandr Holovachov [7], Ann-Marie Waldvogel[8,9] & Philipp H. Schiffer [1,9] ✉

Soil ecosystems are crucial for supporting life, yet little is known about their biodiversity and its distribution in extreme environments. The Atacama Desert, the driest non-polar desert on Earth, has scarce water, high salinity, and metal-rich water bodies, creating challenging conditions for most organisms. Above-ground life has been partially documented, but its soils remain poorly studied. Here we show that soil nematodes, an abundant and diverse group of invertebrates, display distinct biodiversity patterns across the Atacama at multiple biological scales, including genetic, taxonomic, community, and life-cycle levels. Surveys across dune systems, high-altitude mountains, saline lakes, river valleys, and fog oases reveal unique assemblages in each habitat. We find that asexual taxa are more common at higher altitudes, consistent with patterns of geographical parthenogenesis. Genus richness follows a latitudinal gradient and increases with precipitation. These results demonstrate that even in one of the most extreme terrestrial environments, stable soil communities can persist. However, evidence of simplified soil food webs suggests vulnerability to further environmental change. Our findings provide new insights into the mechanisms shaping biodiversity in arid ecosystems and can inform predictions about soil resilience under global climate-driven aridification.

Understanding which factors shape the diversity and distribution of species is a major focus of ecological research. Studies examining the interaction of organisms with their immediate environment focus on the influence of ecological, geological, and geographical parameters in diverse systems such as aquatic and soil habitats[1–5]. From nutrient cycling, mineralization, and pest control[6] to carbon sequestration and storage, soils and their processes play a key role in food security, water supply and quality, and in shaping biodiversity[7]. Globally, at least 58 % of species inhabit soils[8], including microorganisms and different size classes of fauna[9]. Accordingly, soils receive extensive research attention, particularly in arable areas. However, geographical patterns of soil biota in relation to environmental parameters remain poorly understood, largely due to limited data on the true diversity and distribution of many soil organisms[8,10].

Even less information exists for soil systems in extreme environments such as deserts. Abiotic factors such as strong temperature

[1]Department of Biology, University of Cologne, Cologne, Germany. [2]Helmhotlz Center Postdam, GFZ German Research Center for Geosciences, Wissenschaftpark "Albert Einstein", Potsdam, Germany. [3]Leibniz Centre for Tropical Marine Research (ZMT), Bremen, Germany. [4]Faculty of Agriculture, Kasetsart University, Chatuchak, Bangkok, Thailand. [5]Center for Genome Regulation, Facultad de Ciencias, Universidad de Chile, Santiago de Chile, Chile. [6]Centro de Estudios Avanzados en Zonas Aridas, Universidad La Serena, La Serena, Chile. [7]Department of Zoology, Swedish Museum of Natural History, Stockholm, Sweden. [8]School of Life Sciences, Technische Universität München, Iffeldorf, Germany. [9]These authors contributed equally: Ann-Marie Waldvogel, Philipp H. Schiffer. ✉e-mail: lau.villegas@lmu.de; philipp.schiffer@gmail.com

fluctuations, minimal annual precipitation, and high evaporation rates drive desert aridity and pose significant challenges for life[11]. Despite these constraints, deserts and their soils contain spatially distributed microhabitats and niches, including limestone areas, dune systems, and saline patches, that support diverse life forms[11]. With the global trend of aridification[12–14], studying biodiversity in desert soils is essential for predicting organismal distributions under global change.

The Atacama Desert, the driest non-polar desert on Earth, receives less than 200 mm of rainfall annually. Despite the long-term stability of its geological system, regional environmental patterns are apparent[15,16]. Long-term aridity shapes the distribution of sulfates, chlorides, and nitrates in Atacama soils, which in turn affect community composition across taxa[17–20]. Diverse microhabitats in the Atacama include alkaline soils, dune systems, fog oases, and soils with high salinity and arsenic content. These support specialized taxa such as microbial life forms, plants, and a limited number of vertebrates[21,21–31]. Invertebrate biodiversity remains understudied in the Atacama, with research focusing mainly on above-ground fauna[32–34], leaving soil invertebrates poorly sampled and their distribution patterns unclear[8,29,35–37]. Nematodes, one of the most diverse and abundant metazoan groups, contribute to soil nutrient turnover, carbon sequestration, and regulation of bacterial populations, making them important indicators of soil condition[36,38–40]. Their use as functional indicators is based on life cycle characteristics such as feeding type and reproductive strategy[38,41]. Members of Nematoda adapt to extreme conditions, including low oxygen in deep-sea environments[42], daily freezing in Antarctica[43,44], and kilometer-deep mines[45]. Certain families are prevalent and diverse in deserts, surviving low water availability in systems such as the Mojave[46], Chihuahuan[47], Monte[48], and Namib deserts[49]. Many nematode genera also evolve parthenogenetic lineages, hypothesized to have a founder advantage in extreme environments (geographical parthenogenesis)[50,51]. In the Atacama, parthenogenesis may therefore be more frequent than sexual reproduction.

Published records of nematodes from the Atacama are usually based on very few or single individuals[35]. Consequently, it remains unclear whether soil invertebrate biodiversity patterns align with regional and local ecology, adaptive strategies such as feeding type or reproductive mode, or broader global trends such as environmental gradients in species richness[3]. In this study, we analyze the factors shaping invertebrate biodiversity in Atacama soils using nematodes as a model system. We sample six localities across the desert, including sand dunes, saline lake shores, and high-altitude mountain regions, to characterize biogeographic patterns in genetic diversity, life history, and community composition. Despite extreme and persistent aridity, we find strong local patterning across niches such as sand dunes, saline lakes, fog oases, and river valleys. Using genus richness and reproductive mode modeling approaches, we identify annual precipitation and temperature heterogeneity as the main factors associated to genus richness, and found elevation to be the strongest predictor of reproductive mode. We show that even under extreme environmental conditions, soil niches can sustain stable and diverse invertebrate communities, although some areas show signs of simplified food webs.

## Results

In this study, we aimed to analyze different regions of the Atacama and thus established localities and sampling sites in major areas of the desert: (1) the Altiplano area (Tarapacá region) is more humid and known for supporting more vegetation than the hyperarid core of the Atacama[52]; (2) the area around the Quebrada de Aroma (Tarapacá region) has moderately alkaline soils with increasing gypsum as altitude decreases and harbors several microorganism taxa such as Proteobacteria and Actinobacteria[21], (3) the San Francisco area - further on referred to as Eagle Point (Antofagasta region)—is home to several bacterial species with predominant Firmicutes taxa[24], (4) the Salar de

Huasco (Tarapacá region) experiences high UV radiation, salinity and arsenic concentrations supporting several microbial taxa[25,53] as well as three flamingo species, a nearly threatened puma species and several other vertebrates[23], (5) the Paposo area (Antofagasta region) receives water in the form of fog (called camanchaca), it harbors plant species-rich fog oases (known as loma formations)[54,55] and is an area of high endemism[25,56,57] and the (6) Totoral area (Atacama region) is composed of sand dunes where endemic plants of the Apocynaceae family[22], as well as reptiles and birds can be found. In order to analyze patterns of biodiversity in the hyper-arid Atacama Desert using nematodes as a model for soil organisms, we sampled, identified, sequenced, and studied diversity at different levels. We analyzed diversity at the haplotype, genus, family, and community level, including life-history characteristics such as feeding types and reproductive modes. We used linear models as well as a random forest approach to assess the factors that contribute the most to the genus richness and reproductive mode patterns across the desert.

### A rich diversity of roundworms persists in the Atacama

We collected 393 nematode morphotypes from 112 samples collected along six localities throughout the Atacama Desert (Fig. 1B, supplementary Fig. S1). Collected nematodes were then sequenced ("Methods") with a percentage of high quality base pairs above at least 20 (HQ 20 %), 386 sequences from different morphotypes were kept for further analysis. More specifically, we analyzed a total of 39 nematodes from the Altiplano locality, 37 from the Aroma locality, 163 from the Eagle Point locality, 44 from the Paposo locality, 26 from the Salars locality (Salars), and 77 from Totoral dunes locality. Our sampling also included sites where no nematodes could be found (Fig. 1B). These sites are used as a reference for environmental conditions under which nematodes do not seem to persist. We classified the nematodes into 21 families and 36 genera (list of lineages per sample can be found on 10.5281/zenodo.15517342−FamilyGeneraperSample.csv), illustrating a rich diversity of roundworms flourishing in the Atacama. Seven families were found in the Altiplano as well as the Aroma locality, nine families were found in Eagle Point, 12 in the Salars, eight in Totoral Dunes, and ten in Paposo (Fig. 2B). We observed that some generalist families were ubiquitously found, whereas others were geographically restricted. Cephalobidae were present in all localities, whereas representatives of families Panagrolaimidae and Qudsianematidae were found in five of the localities (all except for the Salars). The family Anguinidae was uniquely found in the Eagle Point locality. The families Trichistomatidae and Mermithidae were unique to the Salars locality. Tylencholaimidae was uniquely found in the Paposo locality, and Acuariidae in the Altiplano.

### Genetic diversity and haplotype uniqueness differ between nematode taxa

Given the diversity of retrieved nematode genera and the disparity between ubiquitous and regionally restricted taxa, we wanted to analyze patterns of biogeography on the genetic level. We assessed haplotype clustering and genetic diversity ($\pi$, $\theta$) for seven taxonomic groups using 18S rRNA (Table 1). We studied genetic diversity in two ways: per taxonomic group, including all habitats, and per habitat, analyzing one taxonomic group at a time.

There were between 17 to 166 segregating sites for the different soil nematodes in our networks (supplementary Table S8), and we identified differences in haplotype structure and clustering at the genera and family levels (supplementary Figs. S3–S7). For the genera Acrobeloides, Acrobeles, and Panagrolaimus we found 24, 22, and 20 different haplotypes, respectively (Table 1). For the families included in the order Dorylamida and the superfamily Aphelenchoidea, 16 haplotypes were identified, respectively. The family Plectidae was composed of ten different haplotypes. For Alaimidae, only one haplotype was found (two including the reference sequence) and is shared in three

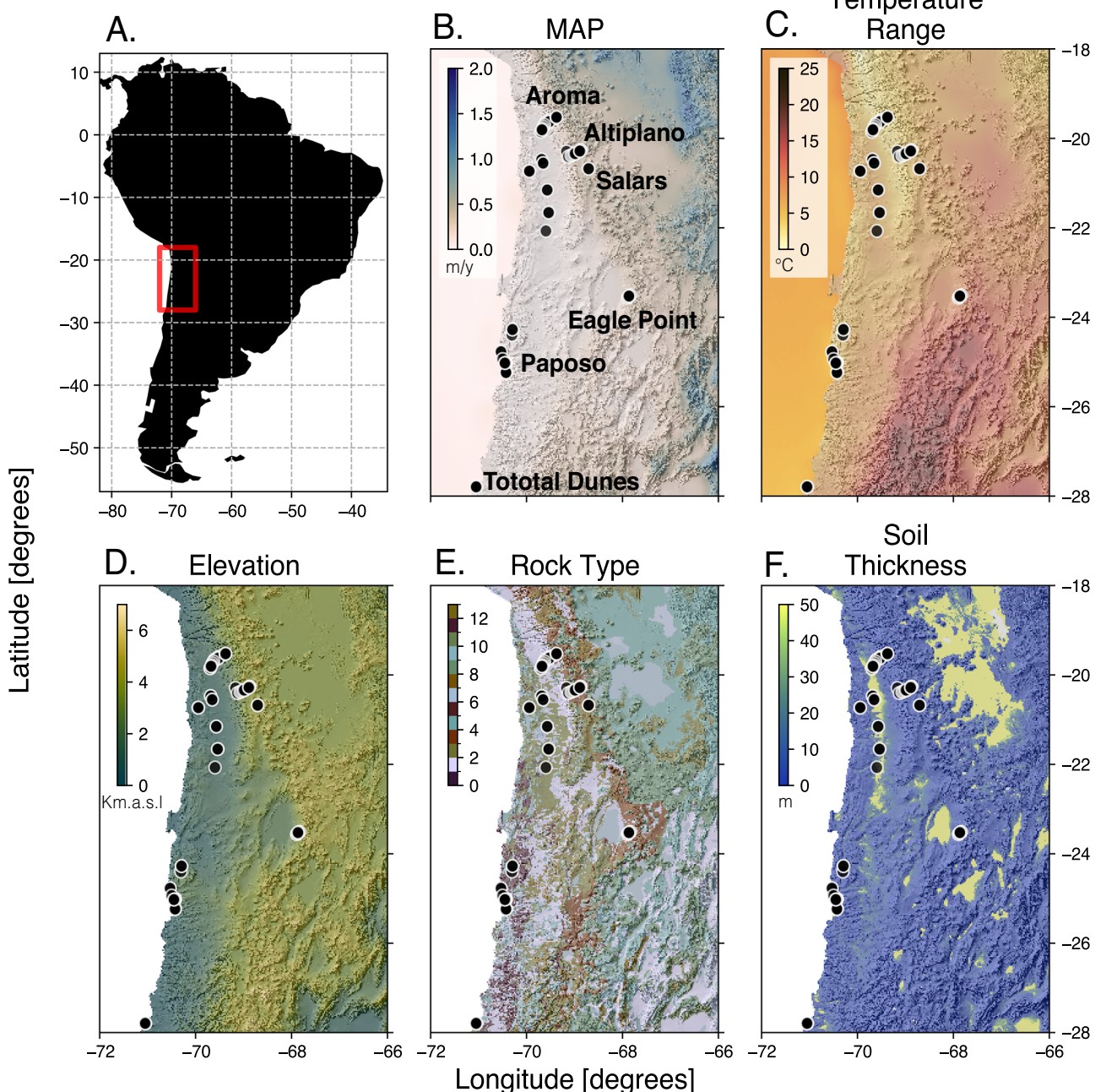

**Fig. 1 | Location and ecological context of the Atacama Desert.** Environmental variables used to model genus richness across the Atacama Desert, with sampling locations shown as points on each map. These maps provide context for the environmental conditions across the study area, including **A** geographical location of the Atacama Desert, **B** mean annual precipitation, **C** temperature range, **D** elevation, **E** rock type (classified following[138], see Supplementary Table S19), and **F** soil thickness. All layers represent spatial data, not model predictions. The maps highlight the environmental gradients and heterogeneity present in the Atacama, which underlie the variation in nematode genus richness observed across the sampled sites.

habitats (Table 1). Overall, we were not able to find any distinct geographical clustering of haplotypes based on the networks, i.e., individual haplotypes were present in geographically distant localities (a complete list of all identified haplotypes can be found on 10.5281/zenodo.15517342). We found genetic diversity at the genus level to be the highest for *Panagrolaimus* ($\pi$ = 0.05788, $\theta$ = 0.04221, Fig. 3A) and the lowest for *Acrobeloides* ($\pi$ = 0.01278, $\theta$ = 0.02192). At the family level, the highest genetic diversity was found for Aphelenchoidea ($\pi$ = 0.11199, $\theta$ = 0.08095, Fig. 3B), and the lowest was found for Plectidae ($\pi$ = 0.00633, $\theta$ = 0.00993).

To better understand the possible effect of geography on genetic distance, we calculated the genetic distance as Euclidean distance and compared it to the geographic distance of the different localities using Mantel statistics based on Spearman's rank correlation ($\rho$). For *Panagrolaimus*, a significant ($p < 0.05$) isolation-by-distance pattern could be found ($r$ = 0.6, $p$ = 0.041667, 119 permutations); no pattern was found for other taxa analysed. With this approach, it appears that geographically distant haplotypes of *Panagrolaimus* are also genetically more distant in their 18S rRNA region. Specifically for this genus, we see the highest diversity in the Eagle Point ($\pi$=0.06548) and the lowest in Totoral Dunes, where only one haplotype was found for several individuals (supplementary Table S9). These results showcase the high variability of genetic diversity within soil nematode taxa in the Atacama Desert and relation to their habitat.

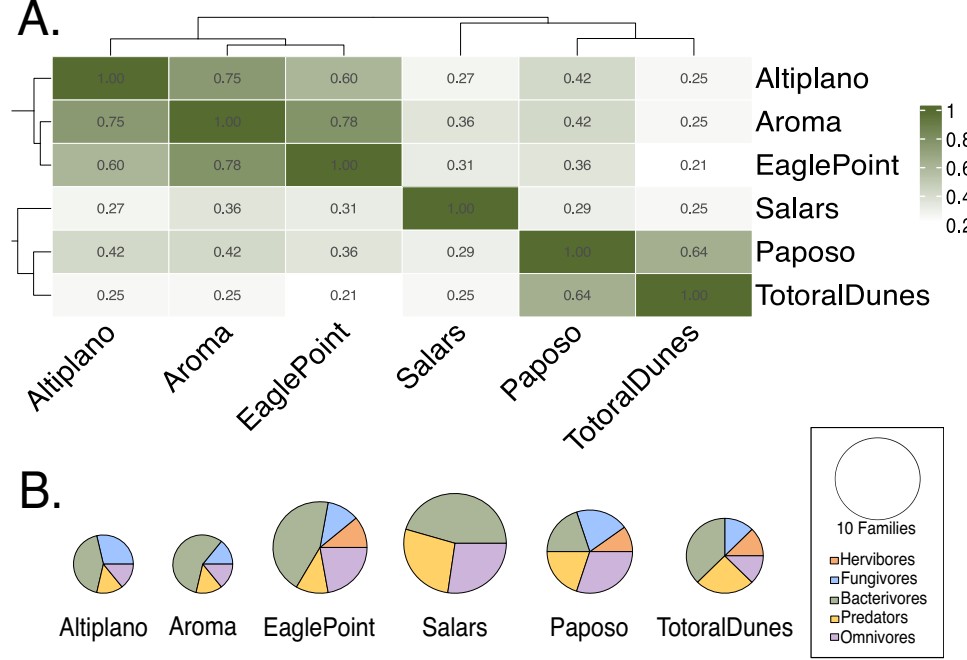

**Fig. 2 | Community composition similarity and feeding type diversity throughout six sampling locations. A** Jaccard similarity index of families found in six localities, intensity of green depicts similarity between locations (darker green indicates higher similarity). **B** Number of families found per locality represented by the size of the circles, the proportion of families found with the different feeding types is represented in the pie chart by the different colors.

**Table 1 | Summary of the different taxonomic groups used to create haplotype networks**

| Taxonomic group | Haplotypes | Total π | Total θ |
|---|---|---|---|
| Plectidae | 10 | 0.00633 | 0.00993 |
| Alaimidae | 2 | 0.01145 | 0.01504 |
| *Acrobeloides* | 24 | 0.01278 | 0.02192 |
| Dorylamida | 16 | 0.01611 | 0.01456 |
| *Acrobeles* | 22 | 0.02877 | 0.02654 |
| *Panagrolaimus* | 20 | 0.05788 | 0.04221 |
| Aphelenchoidea | 16 | 0.11199 | 0.08095 |

The number of haplotypes, total π and total θ are specified, and sorted from low to high π. Genetic diversity is estimated per site.

## The nematode community composition varies between habitats

We assessed soil nematode community composition variation in the six localities employing the Jaccard similarity index at the family level. We found the Aroma and Eagle Point localities to share the highest proportion of families ($J = 0.78$), followed by Altiplano and Aroma ($J = 0.75$). Conversely, we observed the least amount of shared families between Eagle Point and the Totoral Dunes ($J = 0.21$) and the Salars locality when compared to Totoral Dunes ($J = 0.25$)(Fig. 2A). As expected under the isolation-by-distance model, community composition appears to become more dissimilar with increased geographical distance between the different localities (Mantel statistic based on Spearman's rank correlation $\rho$, r 0.5153, $p = 0.048611$, 719 permutations). We were interested to know if a co-occurrence analysis of families would identify a specific geographical pattern, but using 82 family pairs (108 pairs were removed due to unique presence in specific localities) could not retrieve a signal.

## Soils in the Atacama are dominated by different categories of nematodes

We wanted to use nematodes as indicators of soil condition throughout different habitats in the desert by assessing their different functional groups. We categorized nematodes into different feeding types, as well as into the colonizer-persister series based on common life-history characteristics reported in literature and assigned using the NINJA application[41]. Nematodes that are r-strategists (c-p1 colonizer) are considered indicators of resource availability and unstable environments, whereas those that are k-strategists (c-p5 persister) are considered indicators of food web complexity and more stable habitats[58]. We found that the different habitats show a different structure regarding nematode functional groups. Despite the presence of nematodes belonging to all colonizer-persister categories (c-p) across the desert, r-strategists (categories c-p1 and c-p2) are predominant in the Altiplano locality (57.2 % of families found in the locality) as well as in the Aroma and Totoral Dunes localities. Whereas k-strategists (categories c-p4 and c-p5) were the predominant type of families in the Paposo and Salars localities (supplementary Table S11). By assessing the different feeding types of the nematodes analysed, we found that most families in the different habitats are bacterivores and omnivores (41.2% and 20.1% respectively). The remaining nematode families were categorized as predators, fungivores, and herbivores (18.7%, 14.4%, and 5.6% respectively)(Fig. 2B). Families with unicellular eukaryotic feeding type (UEF) were not found in this study (supplementary Table S12). A structuring of the different soil habitats regarding feeding types is apparent, as the Salars only harbor bacterivore, predatory and omnivore families whereas other areas as Totoral Dunes have a more diverse feeding type assemblage with herbivores, fungivores, bacterivores, predators and omnivores present (Fig. 2B).

## Climate and latitude determine soil biodiversity in the Atacama

To understand the driving forces and constraints of nematode taxonomic biodiversity across the Atacama Desert more globally, we used a modeling approach incorporating environmental data. We first aimed to predict genus richness and found that the best predictive model using a probabilistic approach incorporated soil thickness and range of temperature. The model showed the highest

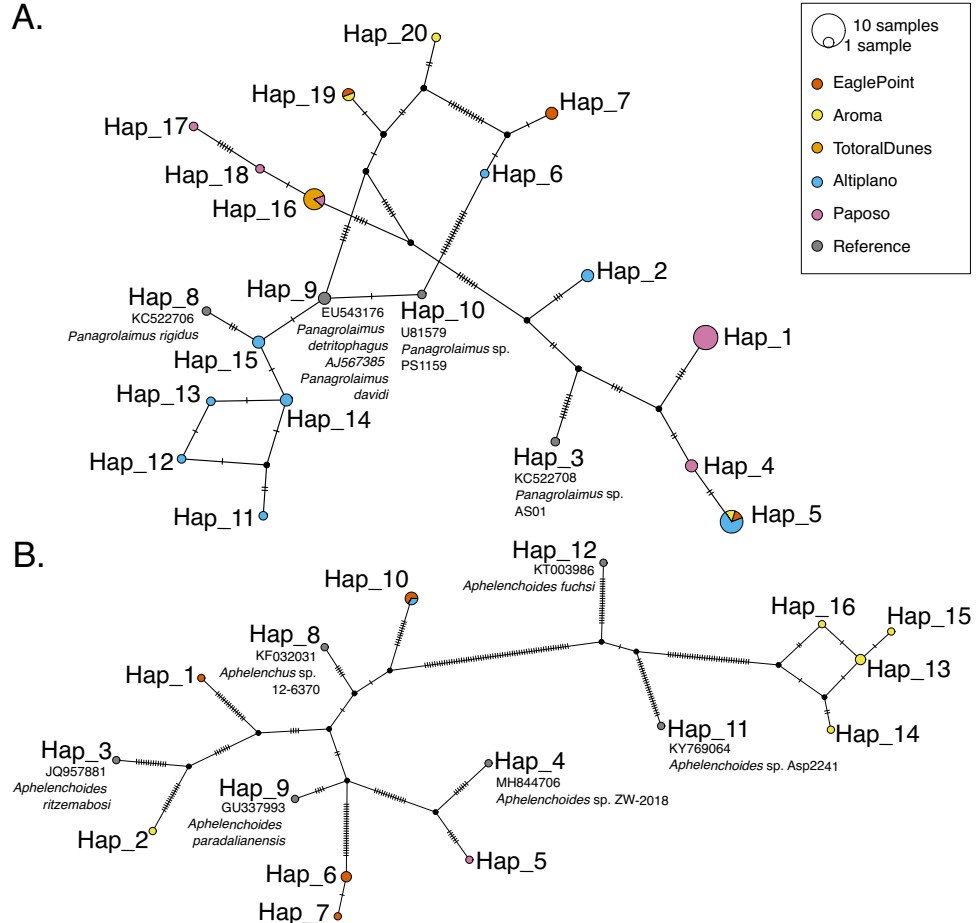

**Fig. 3 | Haplotype networks of nematodes isolated from the desert. A** Haplotype network of *Panagrolaimus* based on 18S rRNA sequences. **B** Haplotype network of Aphelenchoidea based on 18S rRNA sequences. Color refers to sampling locations. Reference sequences were retrieved from the curated 18S-NemaBase[120]. Accession number and species name are written next to the respective node of haplotype (abbreviated to "Hap"). Vertical hatchmarks indicate the number of differences from one node to the next.

## Table 2 | Summary statistics of the linear mixed effect model for genus richness

| Linear mixed effect model for genus richness | | | | |
|---|---|---|---|---|
| **Parameters** | **Value** | **SE** | **t-value** | **p-value** |
| Intercept | −3.296081 | 1.3105763 | −2.514986 | 0.0135* |
| Slope$_{MAP}$ | 0.012731 | 0.0023134 | 5.503008 | 0.0000* |
| Slope$_{rangetemperature}$ | 0.652375 | 0.2329109 | 2.800965 | 0.0062* |
| Random effects: Locality intercept = 0.0086093; Residual = 1.079537 | | | | |

The model incorporates locality as a random effect. Indicated *p*-values are calculated using a two-sided Wald test.

heterogeneity between the different locations. Incorporating locality effects into the models was therefore essential to capture the ecological drivers of genus richness given our dataset. Moreover, we noted that precipitation patterns are not correlated to the distance between localities (Supplementary note 1).

With this approach, we then found that the best predictive model incorporated mean annual precipitation and range of temperature, representing the seasonal thermal amplitude an organism experiences within a given area (temperature heterogeneity). The preferred model showed the highest Akaike weight (0.5263228) and a statistically significant slope for the intercept and both climatic variables (equation 1) (Table 2).

To further validate the results of the tested linear models, we implemented a random forest approach. With this approach, we found that the most important variables for genera richness prediction in our analysis were elevation, mean annual precipitation, latitude, and range of temperature (supplementary Table S17). This corroborates that mean annual precipitation and range of temperature, in congruence with the linear models, have a strong, consistent influence on genus richness prediction. With the three approaches, we found that fewer genera are predicted to occur in the hyper-arid core of the desert (e.g zero for the region of Pampa del Tamarugal), and more genera are predicted to occur, for instance, in the Altiplano region (Fig. 4A). Analyzing the raw data relationship between

Akaike weight (0.4709556) and a statistically significant slope for both environmental variables (supplementary Table S16), which indicates that nematode genera richness in the Atacama Desert could be related to reduced soil thickness and more climate heterogeneity, here represented as the range of temperature. To account for variation across localities in our predictions, we further assessed genus richness using linear mixed-effect models specifying the sampling locality as a random effect. Our analysis confirmed that sampling localities had a significant influence on the modeling process, suggesting that spatial variation plays a crucial role in shaping the observed patterns, potentially due to underlying environmental

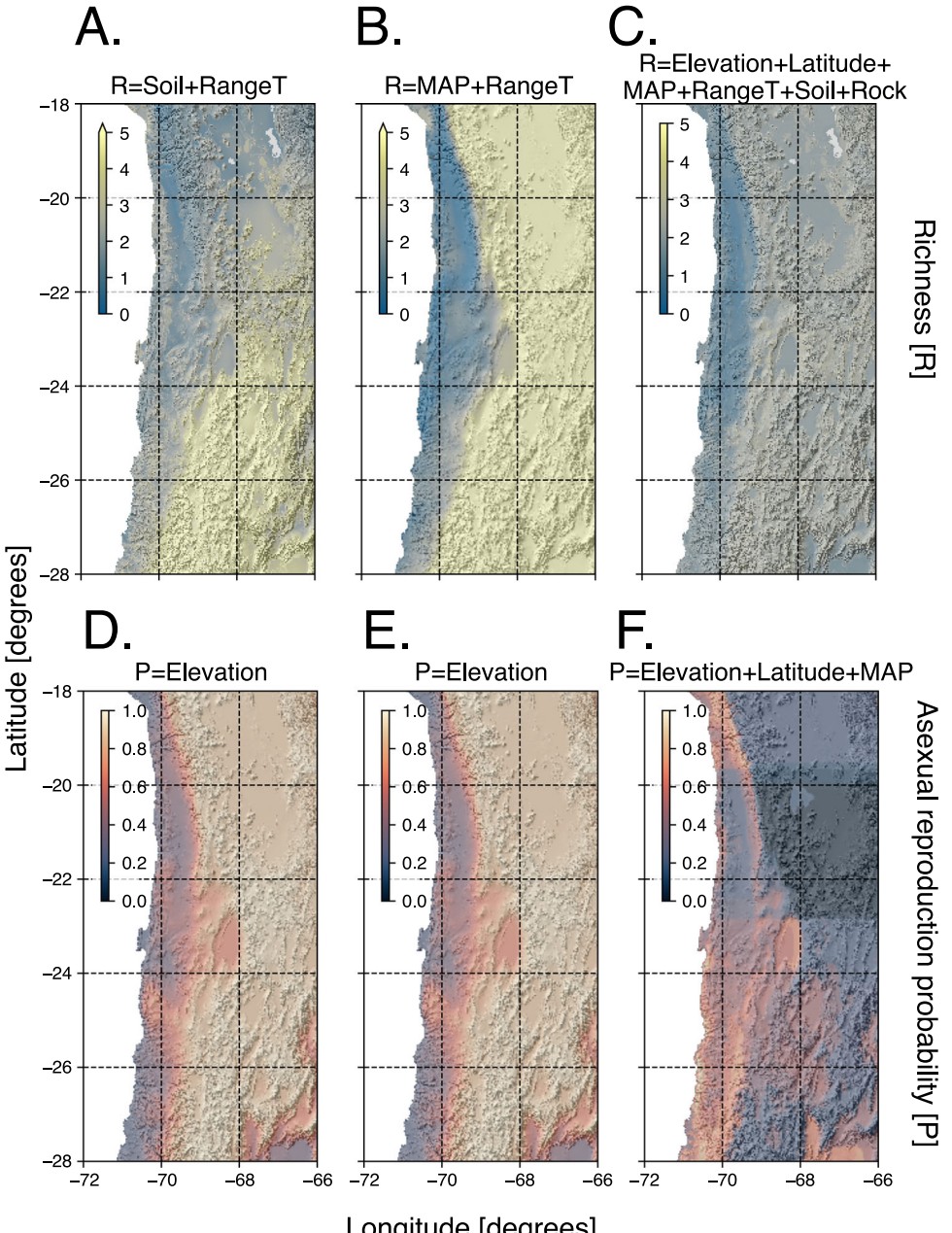

**Fig. 4 | Maps of model predictions for genus richness and the probability of a nematode reproducing asexually in the Atacama Desert.** The upper panel (**A**–**C**) displays predicted genus richness based on different modeling approaches and environmental variables, dark blue indicates lower genera richness and yellow indicates higher genera richness: **A** uses a generalized linear model with soil thickness and temperature range as predictors; **B** applies a linear mixed-effects model based on mean annual precipitation; and **C** employs a random forest model. In these maps, yellow indicates areas with higher predicted genus richness, while blue indicates areas with lower richness. The lower panel (**D**–**F**) presents the predicted probability of asexual reproduction, dark purple indicates higher likelihood of being sexual and light brown indicates higher likelihood of being asexual: **D** and **E** use elevation as a predictor in a generalized linear model and a generalized mixed-effects model, respectively, and **F** shows predictions from a random forest model. In these maps, dark purple indicates areas where sexual reproduction is more likely, while beige shows areas with a higher likelihood of asexual reproduction. "Soil" is used as an abbreviation for soil thickness, "RangeT" as an abbreviation for range of temperature, "MAP" is used as an abbreviation for mean annual precipitation and "Rock" as an abbreviation for rock type.

environmental variables and genus richness, we see that the linear associations were generally weak or non-significant (supplementary Fig. S11), the random forest model captures more complex, potentially nonlinear effects, as shown in the partial dependence plots (supplementary Fig. S12). This highlights that genus richness responds to environmental gradients in a way not fully captured by only linear models. Due to the differing assumptions underlying each modeling approach, a single definitive distribution of genus richness cannot be determined. However, temperature heterogeneity

consistently emerged as a strong predictor across all models. Additionally, mean annual precipitation was identified as a significant factor in both linear mixed-effects and random forest models, underscoring the important role of climate in shaping genus richness.

$$genera\ richness = -3.723925 + 0.015028 * precipitation + 0.688582 * temperature\ range \quad (1)$$

**Table 3 | Summary statistics of the generalized linear mixed effect model for reproductive mode**

| Generalized linear mixed effect model for reproductive mode | | | |
|---|---|---|---|
| **Parameters** | **Value** | **SE** | **z-value** | **p-value** |
| Intercept | −1.2907594 | 0.8585562 | −1.503 | 0.1327 |
| Slope$_{elevation}$ | 0.0008667 | 0.0004082 | 2.123 | 0.0337* |
| Random effects: Locality intercept < 0.001; Residual < 0.001 | | | | |

The model incorporates locality as a random effect. Indicated *p*-values are calculated using a two-sided Wald test.

### Reproductive mode association to environmental variables

The theory of geographical parthenogenesis predicts an excess of asexual taxa in marginal habitats. Given our findings of taxonomically structured nematode communities in different regions of the Atacama Desert, we wanted to test for the specific life-history trait of parthenogenesis (asexual reproduction) using our modeling approach. We established single-worm cultures for a representative subset of our samples. Ten out of the eighteen cultures showed nematodes with parthenogenetic reproduction, i.e., no presence of males and successful propagation from a single unfertilized female. The remaining seven cultures showed sexual reproduction, i.e. presence of males (supplementary Fig. S2G) and no propagation from single females. Sexual strains showed a propagation rate of 0% from single unfertilized female cultures, whereas the asexual strains' propagation rate was 83.3% ( ± 10.48%). We initially tested variables of elevation and latitude as predictors of reproductive mode along with mean annual precipitation and range of temperature, the former two to test for common variables associated with geographical parthenogenesis, the latter two given their effect on genera richness prediction from our linear mixed-effects and random forest models. We tested generalized linear models as well as generalized mixed-effects models to account for the effect of sampling locality (supplementary Tables S20 and S21). In addition, we tested a model including vegetation (supplementary Fig. S14, supplementary Table S20). The best predictive model for reproductive mode incorporated only elevation, showing the highest Akaike weight and a statistically significant slope (supplementary Table S16), while, for example, plant association did not show any effect on the prediction of reproductive mode. We also tested the effect of sampling locality on the prediction of reproductive mode and found, in congruence with the previous approach, that the model incorporating only elevation was preferred (Table 3). Our validation of these linear models with a random forest approach supports that increasing elevation is the main driver for asexual reproductive mode (represented as 1) in soil nematodes in the Atacama (supplementary Table S18). Our models show that the likelihood of nematode taxa being asexual is greater at higher elevations than in lower ones (e.g. coastal area)(Fig. 4). To account for variation across localities in our predictions, we further assessed genus richness using generalized linear mixed effect models specifying the sampling locality as a random effect. In this case, we found that the sampling localities have no significant effect on the likelihood of a reproductive mode.

## Discussion

Altered soil characteristics, along with increasing temperatures, will have an impact on community dynamics and species abundance, especially in times of accelerated global change[59–61]. Dry lands cover around 40% of the Earth's land surface[62]. These areas are projected to increase in aridity as a consequence of climate change[63]. Desertification processes, as an example, soil erosion, acidification, and salinization, have affected over 12% of dry lands already[14], partly due to changing precipitation patterns[64]. Understanding soil biodiversity and species distribution in present-day desert environments could provide

a model for further studies on the effect of desertification and the resilience of soil ecosystems. Using nematodes as a model system, we analyzed which factors shape soil biodiversity in an extreme environment, the Atacama Desert in northern Chile. While aridity generally characterizes the Atacama Desert, we found that nematode diversity shows patterning throughout different niches (e.g., sand dunes, saline lake-shore sediments, fog oases), as well as in relation to ecological and geographical parameters. We identified mean annual precipitation and latitude to be the major predictors of genus richness in the desert and elevation was identified as the primary predictor of reproductive mode. Additionally, the characterization of nematodes as indicators of soil condition, based on their life cycle characteristics, revealed that some niches are stable (taxa that are sensitive to pollutants and disturbance) while others exhibit signs of a poor soil food web with high tolerance to pollutants and disturbance. Here, we describe the biogeographic patterns of nematodes, focusing on their diversity, life-history, and community composition in the Atacama Desert, where studies on soil organisms, such as roundworms, are still scarce[29,35].

### Atacama soils harbor a rich nematode diversity

We analyzed diversity from the haplotype level (genetic diversity) to community level patterns (genera richness and Jaccard similarity at the family level). We report that the Atacama Desert is home to at least 36 different genera of nematodes grouped in 21 families, spread across various habitats. We identified cosmopolitan nematode families, including members of Cephalobidae and Panagrolaimidae, which have been documented in other desert environments such as the Namib Desert[65,66], the Judean Desert[67], the Mojave Desert[68,69], and the Monte Desert[70]. These families have also been reported in the arid region of Arica in the Atacama Desert[37]. We found both families in most analyzed areas and aimed to assess the genetic differences among individuals from different localities and their relationship to ecological, geological, and geographical factors. Particularly for the family Panagrolaimidae, all sequenced individuals belonged to only one genus (*Panagrolaimus*), which displayed an isolation-by-distance pattern (IBD). We were able to detect this pattern in *Panagrolaimus* specifically, as we found a high enough proportion of sexually reproducing individuals present in our samples (asexually reproducing species do not have gene flow and thus by default show separation). A possible explanation for IBD is limited dispersal, which has been reported to be lower in nematodes of this genus when compared to other species in a laboratory setting[71]. Generally, dispersal is lower for below-ground organisms when compared to above-ground taxa[72]. Dispersal can be mediated by wind in small invertebrates like nematodes[73], but the effect of wind dispersal has not been studied to date in the Atacama.

### Soil biodiversity in the Atacama is highly structured and in parts sensitive to disturbance

By examining communities at both the haplotype and family levels across different desert niches, we identified distinct local patterns. For example, the Jaccard similarity analysis revealed distinct communities in salars compared to other regions. The influx of freshwater from complex underground aquifers into high-altitude wetlands, along with elements like lithium and arsenic, high salt concentrations, and seasonal fluctuations, contributes to the formation of these unique ecosystems[53,74,75]. Microbial life, proposed as the dominant life form in this area, displays density and structure changes through dry and wet seasons[75], which in consequence can have an effect on the nematode community in the area[76], given that most families found in the wetlands were bacterivores. Differences in the taxa found in the different areas can also provide information on the complexity of the soil food web; both the salars and Paposo were dominated by persisters that are described to be more sensitive to environmental disturbance and often have a narrow ecological amplitude. Even if these indicators are developed in communities of agricultural soils[77], which are expected to

differ significantly from the oligotrohpic desert soils, the relative difference in the proportion of persisters among the here investigated localities ( ≈ 10% more than colonizers), highlights the sensitivity of food webs in the salars and Paposo. This aligns with observed and reported threats to these ecological niches that are mining, plant extraction, pasture beyond the area's capacity, truck transit with dangerous materials, uncontrolled tourism, and habitat loss due to vehicle transit off roads[78].

In contrast, in the Altiplano, the Totoral Dunes, and the Aroma area the dominant groups were colonizers that are described to be more resistant to adverse environmental conditions in agricultural soils, indicating poor soil food web function in the absence of some feeding types[79,80]. Even if the here investigated soils cannot compare to eutrophic agricultural conditions, the relative dominance in colonizers in comparison to the soils from the Salar and Paposo localities indicate altered food web function in Apltiplano, Totoral Dunes, and Aroma. Sand dune systems, such as Totoral Dunes, have already been reported to be dominated by coloniser taxa due to their broad ecological range, with high abundances of the family Cephalobidae throughout coastal sand dunes[81,82].

## Soil community diversity patterns across the Atacama Desert are primarily shaped by climate

To further understand if broader scale patterns could be found for the soil communities, we implemented a genera richness modeling approach. In isolated areas such as the Atacama Desert, comprehensive field studies that encompass the entire area are challenging due to the desert's complex topography and large terrain extension. Modeling approaches provide a simplified representation of real-world phenomena[83] specifically for multifaceted scenarios, as we observed in the Atacama. These models are thus an excellent tool to disentangle the driving factors of species and community distributions by incorporating the complex interplay of ecological, geographical and edaphic factors[84].

Our results highlight the key role of climate, particularly mean annual precipitation and temperature heterogeneity, in shaping nematode genera richness. Precipitation, as a proxy for water availability, is a well-established driver of soil biodiversity across various taxa, including protists, bacteria, fungi, arachnids, rotifers, and nematodes[85]. In desert environments, higher precipitation can enhance biodiversity by improving nutrient availability and soil moisture[86]. Specifically, in the Atacama Desert, it has been linked to species distribution and habitat suitability in beetles[87] and increased microbial diversity at higher elevations[21].

Soil nematodes, like other invertebrates[88], rely on the thin water film between soil particles for movement, making water availability crucial[89]. While some can survive dry conditions through anhydrobiosis[90–92], active nematodes depend on precipitation to sustain mobility and biodiversity. Increased precipitation has also been linked to shifts in nematode communities, particularly an increase in bacterial-feeding nematodes, likely due to a rise in bacterial biomass[93].

Beyond precipitation, we found that temperature heterogeneity (seasonal temperature variation) also influences genera richness in the Atacama Desert. While temperature fluctuations are known to shape community dynamics, their precise role in food web interactions remains incompletely understood[94]. Temperature heterogeneity has been identified as a key factor in population structure[95], a driver of increased protist densities[96], and a contributor to significant community dissimilarities[97]. Variations in climate across localities may create distinct micro-niches, fostering the establishment of diverse biological communities.

Although not identified as the primary drivers of nematode diversity in our study, latitude, elevation, and edaphic factors (such as soil thickness and rock type) may still influence community diversity in the extreme environment of the Atacama Desert. These edaphic factors have been proposed as key regulators of soil biodiversity by shaping various environmental conditions, including weathering processes, soil pH, clay formation, water infiltration, and retention capacity[98,99]. Plant coverage and type can also play an important role in supporting nematode diversity, as plants have been found to be biodiversity hotspots in desert ecosystems for both microorganisms (that serve as food source for nematodes) and invertebrates[49,100]. However, fine scale analysis of soil and plant properties is not included this study and should be consider in future work.

Latitude, widely recognized as a major determinant of taxonomic diversity, is central to the concept of the latitudinal diversity gradient (LDG)[101,102]. While the LDG has been empirically observed across numerous taxa[103–105], its underlying mechanisms remain unclear. Latitude often interacts with other ecological and climatic factors, making it difficult to disentangle its direct effects from confounding influences[3,106–108].

While climate appears to be the primary driver of nematode biodiversity in the Atacama Desert, it is important to acknowledge that this conclusion is based on the current dataset and represents an initial assessment of potential taxa distribution across this complex landscape. A more comprehensive understanding of biodiversity in the region will require systematic sampling with an increased sample size, particularly capturing diversity in high-altitude and southern areas of the desert that remain difficult to access.

Moreover, the broad patterns identified through different modeling approaches do not account for fine-scale habitat heterogeneity (e.g., water bodies, specific plant associations, or areas with persistent shade), which likely contribute to the availability of distinct ecological niches for nematodes. Despite these limitations, our analysis provides a valuable framework for understanding the potential distribution of nematode taxa in this extreme environment. By offering an initial biogeographic perspective, this study serves as a foundation for future research and targeted field experiments, ultimately enhancing our knowledge of nematode biodiversity in the Atacama Desert.

## Asexual nematodes are more prevalent in higher altitudes in the Atacama

The hypothesis of geographical parthenogenesis posits that at high latitudes, high altitudes and in extreme environments, asexually reproducing taxa are more prevalent when compared to sexually reproducing ones[109–113]. Testing for this hypothesis allowed us to investigate how the relationship between taxa richness and complex biological traits (reproduction) evolved within some nematodes under globally extreme environmental conditions, finding that with increasing elevation, soil nematodes in the Atacama are more likely to reproduce asexually than sexually is supported by the Red Queen Hypothesis (RQH). The RQH suggests that in environments with high biotic interactions, sexual taxa are at advantage, while in harsh conditions with lower interactions, such as high altitudes, asexually reproducing taxa are more prevalent[114,115]. While our findings appear to indicate that even in an environment with generally harsh (or extreme) conditions, as the Atacama Desert, parthenogenetic taxa are most successful at the margins, we note that larger sampling sizes across multiple taxa will be needed for our understanding of geographical parthenogenesis as a phenomenon in nature. At the very least our data on the reproductive mode of soil nematodes demonstrates that the evolution of biological traits is influenced by ecological, edaphic and geographic conditions in very stable environments, like the Atacama. On a large time-scale the Atacama shows a remarkable climatic stability (aridity for over 150 Ma), showing no significant latitudinal movement since the late Jurassic[15]. However, on shorter time-scales climatic variation is present, our results reveal nematode diversity across different levels of biological organization but do not provide insights into abundance, that eventually can be highly variable across seasons (e.g., during the Altiplano rainfall season). Nevertheless, we expect that

taxonomic diversity remains relatively stable throughout the year for the localities and sampling points analyzed, given that several taxa are capable of undergoing anhydrobiosis.

In summary, soils of the Atacama Desert are home to a diverse fauna of nematodes. While ecological strategies follow regional-local patterns (e.g colonizer-persister distribution), reproductive strategies and genera richness follow a global trend. Even in a system that is seen as an extreme on the environmental scale, diverse communities are distributed along the desert with more genera rich communities in areas with higher precipitation and lower latitudes. Remarkably, asexual taxa appear to be more prominent in the marginal ranges of high altitudes. Future research should prioritize systematic sampling to evaluate the robustness of the model-derived predictions we present in this study.

Our study highlights the necessity of biodiversity and ecosystem health assessments to understand the territory's biological uniqueness. The Atacama has been historically explored for its notable mineral richness; consequently, mining has been a common activity throughout the desert, threatening the stability of unique ecosystems and the communities that protect and manage these territories[116]. Focusing on biodiversity research in the Atacama Desert would not only guide our understanding of biological evolution but also our understanding of the limits to life under global change.

## Methods
### Sampling, extraction and morphological identification
All research conducted in this study complies with relevant ethical regulations. Fieldwork and sampling were carried out under the required permits and permissions, including authorization from the Corporación Nacional Forestal (CONAF; Permit 09/2023) for work in Salar de Huasco. The study was conducted in full cooperation with local communities, including the Comunidad Lickan Antay de Toconao and the Comunidad Indígena Aymara Laguna del Huasco, who granted access to their territories. Local researchers were actively involved in the research process. A field school was conducted in Collaboration with Chilean researchers, university and school students where training on sampling methods, molecular work and data analysis was conducted. Additionally, reports on the results obtained were provided to the Corporación Nacional Forestal in Spanish as soon as the results were obtained. The research was designed to be locally relevant, addressing environmental and conservation questions of importance to the communities. Local and regional literature and prior studies conducted in the area were carefully considered and cited to ensure contextual relevance.

Samples were taken in six different localities across the desert. Here we define a locality as an environmentally distinct area, identified based on visible ecological and landscape features. We determined them based on direct field observation and previously defined priority sites in ref. 117, considering factors such as dominant vegetation, presence of water bodies and landforms while considering habitat suitability for nematodes. We named the localities as Altiplano (ALT), Aroma (ARO), Eagle Point (EPT), Salar de Huasco and Laguna Grande (Salars), Paposo (PAP) and Totoral Dunes (TDT). Sampling in the protected Salar de Huasco was conducted under the authorization (09/2023) of the Corporación Nacional Forestal (CONAF). In this study, 7 samples from Altiplano, 11 samples from Salars, 15 samples from Paposo, 20 samples from Totoral Dunes, 23 samples from Aroma and 21 samples from the Eagle point were analyzed (supplementary Tables S1–S6). Other areas sampled where no nematodes were found and are not part of the defined localities are later on included in the modeling process (supplementary Table S7).

Soil samples of approximately 500g were taken in the different locations using a shovel, from the upper 0–10 cm and up to 30 cm of each site and stored in zip-lock bags. Samples taken near to plants corresponded to sediment ~10 cm away from the plant stem, or

from plant litter in the case of Paposo. They were initially checked for the presence/absence of nematodes on-site by extracting a portion from the soil samples using seeding trays with one layer Kimberly-Clark Kimtech science precision wipes 7551. Trays were flooded from the bottom and samples were allowed to re-hydrate overnight. The water was then filtered through sieves of 80 and 120 μm. Nematodes were collected on petri dishes and examined using a Zeiss Stemi 2000 microscope; this procedure was also followed in a laboratory setting for further morphological and molecular work.

Adult nematodes were morphologically identified using a Zeiss Axioplan 2 light microscope and further examined at 10X, 20X and 40X following multiple diagnostic features as shown in supplementary Fig. S2. Examined individuals were categorized as morphotypes to account for morphological variation within nematode families. Morphotype representatives for each family found on the samples were stored in 10μl of nuclease-free water and stored at -20 degrees Centigrade for further processing.

### DNA extraction, barcoding and sequence processing
DNA extraction was performed on whole single individual nematodes using a modified version of the HotSHOT DNA extraction method[118]. PCR was performed using the Cytiva PuReTaq Ready-To-Go™ PCR Beads or REDTaq® ReadyMix™ PCR-mix. The genetic marker 18S rRNA was amplified using the forward (CGCGAATRGCTCATTACAACAGC) and reverse (GGGCGGTATCTGATCGCC) primers described in ref. 119, targeting the V2, V3 and V4 regions. The thermocycling conditions used were: denaturation at 94 °C for 5 min, 35 cycles of amplification (94 °C for 30 s; 54 °C for 30 s; 72 °C for 70 s), followed by a final extension at 72 °C for 5 min. Gel electrophoresis was performed (1.5% gel) to assess the amplicons. PCR products were purified using the Exo-CIP™ Rapid PCR Cleanup Kit (NEB - E1050L). DNA samples were submitted for sequencing to Eurofins Genomics and Genewiz.

Sequences were analyzed using Geneious Prime® (v. 2024.0.5). Only sequences with at least 20% high-quality bases (HQ%) were kept for further processing. Sequences were trimmed at the ends (regions with more than 1% chance of an error per base) and sequence similarity was analyzed using BLAST+ (v. 2.12.0) implementing the curated 18S-NemaBase[120] using the blastn function incorporated in Geneious Prime. Best hits were assigned according to the lowest E-value and a high percentage of identity (list with accession numbers and sample information can be found on 10.5281/zenodo.15517342).

### Genetic diversity and haplotype uniqueness
Nematode families represented by five or more 18s rRNA sequences from different samples and present in more than one locality were used to generate haplotype networks. Sequences were grouped based on their sequence similarity to reference sequences of the curated 18S-NemaBase database[120], identified to family and if possible genus level. Due to lack of resolution in the sequences, some haplotype networks were created based on family level instead of genus level. The grouped sequences were aligned (multiple alignments) using Clustal Omega (v. 1.2.3). Short sequences (<300 bp) or with lack of overlap were removed. Alignments were manually trimmed in Geneious prime according to sequence overlap. Trimmed and curated alignments were used as input for DnaSP (v. 6.12.03)[121] to calculate nucleotide diversity ($\pi$) and the Watterson's estimator ($\theta$). Haplotype data files were created in DnaSP and according to this trait matrices defining the occurrences of the different haplotypes were generated and then used as input for PopART (v. 1.7)[122]. Haplotype networks were constructed using the TCS inference method[123]. Networks were finalized using Inkscape (v. 1.3.2). Sequence sets were defined in DnaSP to estimate $\pi$ for each location where enough sequences were present (alignments, haplotype matrix and final output can be found on https://doi.org/10.5281/zenodo.15517342).

## Community composition throughout the Atacama

We estimated the Jaccard disimilarity index ($\beta$ diversity) using presence/absence data of families found in 6 localities (Altiplano, Aroma, Eagle Point, Salars, Paposo and Totoral Dunes), here we analyzed aggregates of communities per locality and do not discriminate by soil sample (bag). We used the vegan package (v 2.6-4)[124] and visualized the results inverted (Jaccard similarity) as a heatmap using pheatmap (v. 1.0.12)[125] in R. We further on analyzed family co-occurrence using the cooccur function of the cooccur package (v. 1.3)[126]. To test for isolation-by-distance patterns, we estimated distance between all combinations of sampling localities on centroids using the sf (v. 1.0-16)[127,128] and geodist (v. 0.1.0)[129] packages. Following, a mantel test was performed comparing Jaccard dissimilarity to geographical distance using the Mantel function of the vegan package with the Spearman's rank correlation $\rho$. The nucleotide diversity for each location of the different groups was also tested for isolation-by-distance patterns using the same approach. Nucleotide diversity was transformed to genetic distance by estimating the Euclidean distance using the base R package stats[130].

## Nematodes as soil indicators

Presence-absence information at the Family level for each locality was used as input for the application NINJA: Nematode INdicator Joint Analysis[41]. We assessed the coloniser-persister structure assemblage of soil nematode families found in the Atacama Desert as well as the feeding type composition in each of the sampling localities.

## Modeling ecological constraints of nematode biodiversity

We evaluated climatic (environmental, e.g. precipitation) and geomorphological (edaphic and geographic, e,g. topographic complexity and latitude respectively) variables as predictors for genera richness in the Atacama Desert. We tested 57 probabilistic models using generalized least squares linear regression from the package nlme (v. 3.1)[131,132] on R and 57 linear mixed effect models using lme from the package nlme, using the sampling localities as a random effect. Data on genera presence in the Atacama Desert was obtained as described through the morphological and genetic identification of the nematodes. The geomorphological variables evaluated were elevation, topographic complexity, curvature, soil thickness, rock type and latitude (supplementary Table S13). We obtained elevation data from a digital elevation model at 15 arc sec resolution[133] and calculated a) topographic complexity as the standard deviation of elevation in a coarser 0.01 degree resolution grid[134] and b) curvature as the second derivative of elevation with respect to the aspect in a 3 × 3 grid[135] as implemented in the RichDEM package (v. 0.0.3)[136] in Python. Soil thickness and rock-type data were obtained from publicly available datasets compiled respectively by Pelletier et al.,[137] and Hartmann et al.,[138]. The evaluated climatic variables were mean annual temperature, mean annual precipitation, temperature range, and precipitation range. Here, 'range' is defined as the difference between seasonal (rather than daily) minimum and maximum values, serving as a proxy for climatic heterogeneity. Temperature and precipitation data were obtained from the openly accessible Chelsea version 2 climate dataset[139,140]. A correlation matrix plot was obtained using the cor and corrplot functions from the corrplot package (v. 0.92)[141] to assess variable correlations ( > |0.65|) (Supplementary Fig. S9), only a representative variable from highly correlated pairs was included in the analysis to avoid redundancy. We furhter conducted a principle component analysis using the prcomp function of the stats package (v. 4.3.2)[130], the PCA was visualized using the factoextra package (v. 1.0.7)[142] in R. We log-transformed soil thickness (supplementary Fig. S8) to fulfill model assumptions. Bivariate relationships between genus richness and each numeric environmental predictor were explored with faceted scatter-plots to obtain the coefficient of

determination ($R^2$) using broom (v. 1.0.7)[143]. Bioinformatic pipelines were optimized using ChatGPT.

We evaluated a total of 114 models, including null models (Supplementary Tables S14 and S15), which incorporated various combinations of selected predictive variables. Half of the tested models were least squares models (GLS), while the other half accounted for the same variables but included localities as a random effect to control for influence of the localities (linear mixed effect models–LME). The variables were selected based on two criteria: 1) variables should not display strong correlation with each other, and 2) the variable contribution to explain the variability along the first and second principal components must be high and among the top variables (supplementary Fig. S10 and S13). Furthermore, the residuals of each model fit were visually evaluated and a variance structure adjustment was performed where necessary following[144]. The preferred predictive models per approach (GLS and LME) were then selected based on the Bayesian information criterion (BIC) and further estimation of Akaike weights (wm)[145], as the latter provides the probability that a model is the best model according to the set of models tested and the experimental data[146]. The preferred predictive models were then used to obtain a map representing the possible nematode diversity in terms of genera richness. We further on tested variable importance of the previously used variable set using the machine learning algorithm Random Forest using the package randomForest (v. 4.7-1.1)[147] to validate the results obtained in the linear models. Training and testing set were created using the rsample (v. 1.2.1). The model was then evaluated using caret (v. 6.0-94). Partial dependence plots (PDPs) were created to visualize the modeled effects of key environmental variables on predicted genus richness using the fitted random forest model using the pdp package (v. 0.8.2)[148].

## Reproductive mode determination and association to environmental variables

Laboratory cultures were established from nematodes extracted from the soil samples. Worms were propagated on nematode growth medium (NGM) plates containing E. coli strain OP50 as a food source. Single-strain established cultures were analyzed morphologically to identify males and females of a strain. Cultures with only females were categorized as possible parthenogens, and cultures with both males and females present were categorized as possible gonochoristic. In addition, single juvenile stage nematode cultures were propagated (to ensure no mating) to determine the reproductive mode of the strains as described in ref. 149. Between 5 to 30 single juvenile cultures were set up for reproductive mode corroboration per strain. We consider asexual cultures where larvae grew into adults and laid eggs without fertilization from a male.

Models testing elevation, precipitation and mean annual precipitation as predictors for reproductive mode (sexual or asexual). Environmental data was collected as in the previous section. Two model types were employed: (a) a logistic regression analysis using the glm function from the stats package in R (v. 4.3.2), with a binomial family distribution (*family = binomial*), and (b) a generalized linear mixed-effects model, implemented with glmer from the lme4 package (v. 1.1-36), incorporating localities as a random effect to account for spatial variability in the samples. Following, additional models were tested including vegetation coverage and percentage of phanerophytes estimated for the samples where reproductive mode could be assessed. Plant identification was done through morphological identification. We further checked which variables were most important by using a Random Forest model with the randomForest package (v. 4.7-1.1)[147], to confirm the findings from the logistic regression and linear mixed-effect model. The data was split into training and testing sets with rsample (v. 1.2.1), and the model's performance was then tested using caret (v. 6.0-94).

 

## Reporting summary

Further information on research design is available in the Nature Portfolio Reporting Summary linked to this article.

## Data availability

The sequencing data generated in this study are deposited on Gen-Bank under accession numbers PQ587582 [https://www.ncbi.nlm.nih.gov/nuccore/PQ587582]–PQ587967 [https://www.ncbi.nlm.nih.gov/nuccore/PQ587967]. Accession numbers per sample can also be found in the Zenodo repository [https://doi.org/10.5281/zenodo.13880073] within the file *List_of_Families_Genera_Haplotypes_with_accession_20250225.csv*.

## Code availability

The code implemented in this study along with the data required to run the code can be found on GitHub https://github.com/lauraivillegasr/BiogeographyDesertand is also available as a reproducible run on CodeOcean under the https://doi.org/10.24433/CO.6395549.v3.

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

## Acknowledgements

We would like to thank Julian Pfeiffer, Siwen Ding, Julia Caro-Valenzuela, Thalia Sotiropoulos for their support. We would like to thank the Comunidad Lickan Antay de Toconao and the Comunidad Indígena Aymara Laguna del Huasco for allowing us in their territory, as well as the Corporación Nacional Forestal (CONAF) for providing the authorization (09/2023) for Salar de Huasco. We would like to highlight the importance of collaborating with local Indigenous communities. This allows researchers to learn from their ancestral knowledge on the territories that have been under their management and protection for generations. This ancestral knowledge enriches scientific understanding, and in turn, the data produced can inform and enhance the management of their territories. Such an approach ensures high-quality scientific output that can reach beyond the academic community and have a direct impact on society. This research is part of the DFG funded collaborative research centre CRC1211 (Earth - Evolution at the Dry Limit) and was conducted within suproject B08 [grant number 268236062] awarded to A-M.W. and P.H.S., and the grant ANID/Milenio/ICN2021_044 awarded to M.L.A. The DFG ENP grant (434028868) funded P.H.S.

## Author contributions

O.H., A-M.W. and P.H.S. designed the research. M.L.A. facilitated sampling and provided resources. L.V.,L.P., N.W. and A.Su. performed the research. L.V., L.P., A.St. and E.A-T. analyzed the data. L.V., L.P., O.H, A-M.W. and P.H.S. wrote the paper.

## Funding

## Competing interests

The authors declare no competing interests.
