## [Transparent Peer Review file · Nature Communications]

Geographic distribution of nematodes in the Atacama is associated with elevation, climate gradients and parthenogenesis

Corresponding Author: Dr Laura Villegas

Version 0:

Reviewer comments:

Reviewer #1

(Remarks to the Author)

The authors conducted a survey of nematode communities in 231 soils from six different habitats from the Atacama Desert in Chile. They employed several techniques to gain ecological insights from this survey. First, they characterized the diversity (morphospecies/family/genus) of the communities through morphological analyses, augmented by partial 18S barcoding (haplotypes). Second, they modelled the structure of these communities incorporating several variables, including latitude, elevation, precipitation and temperature. Finally, they studied the reproductive habits of nematode that they were able to culture and correlated that to environmental characteristics.

Key results:

1. Characterization of the nematode taxa present in Atacama Desert soil – this is new information as no work has been done on these nematodes that I know of for over 50 years.
2. Assessment of B-diversity (Jaccard Index) through incorporation of different sites with different characteristics (latitude, elevation, MAP)
3. Finding a relationship between geographic and genetic distance for the genus *Panagrolaimus* through evaluation of biogeography and 18S genetic diversity of select taxa.
4. Identification of positive effect of a latitudinal diversity gradient and increasing precipitation.
5. A preliminary finding that asexual reproduction is positively correlated to elevation.

Validity:

In so far as key results 1 & 2, the expertise of the team and the combination of morphological and barcoding approaches to identify the nematodes means the data is valid.

I think that use of the Bongers Index to characterize “poor soil food webs” that are “sensitive to disturbance” is a big stretch. The Index was developed mainly for agricultural systems and to investigate the impact of disturbance in those systems. I think, in the context of this study, the Index has little utility. It’s trying to impose what is known about the ecology of nematodes from eutrophic systems to the oligotrophic desert, and I don’t believe it is valid or necessary. Are any of these sites being actively disturbed in some manner? If not, saying the nematode communities are “sensitive to disturbance” isn’t really valid. While lower cp nematodes might be over-represented in a disturbed agricultural soil, their presence in a desert soils is due to different phenomena.

Regarding key result 2, I think the modelling effort is interesting, but I found that some of the environmental data used was strange (see below). Discussion of how the soil habitat changes with the variables that were important was absent (eg, how do MAP, latitude, and elevation actually influence soils as a habitat for nematodes).

Regarding 3: The sequencing effort seems valid and sufficient to be able to examine the haplotype diversity.

Regarding 4: I think this is very, very interesting data, and I appreciate that the authors were candid in the paper regarding the limited sample size. I think saying in the abstract that the distribution of “asexual taxa appears to be in the marginal ranges of high altitudes” is not valid. As far as I can tell, you found a positive correlation with elevation for the small number of cultures established. The data does not show that asexual taxa are not found at lower elevations.

Significance:

Very few studies have attempted to examine nematode genetic diversity within taxa across sites, as the authors have done here. As well, very little work has been done to connect nematode reproductive strategy to the environment. So, these are good questions, and this is novel work of significance to the soil nematode community.

Data and methodology:

I think everything seems clear and appropriate.

Analytical approach:

Regarding the modelling effort to investigate environmental variables and nematode diversity: I think some edaphic and vegetation-related variables would have been useful in this effort, and I wondered if some of this data was collected. For example, soil properties (organic C is very important to nematodes) or NDVI. “Plant association” was mentioned at Line 228 as a variable, vegetation coverage and other variables are in Table S11. For me, the inclusion of Rock type and Soil thickness were not justified beyond being publicly available data (and indeed, these variables did not prove important in any of the models – I would have not put them in the paper at all – maybe leave them out of the figure, as done with topographic complexity and curvature, Line 416). To me, I think the significance of latitude, MAP and elevation on nematode communities is likely due to impacts on vegetation and thus soil organic matter (overall soil habitability), and I think these mechanisms should be at least discussed in the paper. I appreciate not wanting to include codependent variables in the modeling effort, however (but aren’t latitude and precipitation codependent? Surprising not, per S9).

Suggested improvements:

What is a “transect” in this study? It seems that samples were collected haphazardly within sites, but “transect” implies some scheme that was not described in the paper. I would perhaps not describe the sampling as along a transect.

Assuming the number was different for each site, can you provide a range for the number of samples taken? (Line 345) Also, can you clarify the degree of distance from plants that the samples were collected? Were you under the canopy? Was their litter?

I think that addition of a table in the paper or supplement that is a full list of the nematode family/genera and number of haplotypes found at each site is essential and would increase the utility of this work.

I would leave off the Bongers analysis – I don’t think it is informative.

Clarity and context:

I have made some comments above as to where I think more context would be helpful (interpretation of modelling results).

In Figure A: What is “temperature range”? Is that yearly? This is just not a variable I am familiar with.

Lines 153-154: I have read this many times and I cannot make sense of this text, but I feel like maybe it’s important.

Figure 3: Remove UEF from legend – presence is confusing and unnecessary.

References

The paper cites the literature appropriately.

My expertise: I feel comfortable reviewing all aspects of the work.

(Remarks on code availability)

Reviewer #2

(Remarks to the Author)

This manuscript presents a comprehensive investigation of nematode diversity in various subregions of the Atacama. Diversity (genetic, taxonomic, reproductive strategies, functional groups) are investigated and statistical approaches used to determine best predictors (elevation, MAP) of those patterns. This is a unique environment, with limited investigations of the soil biodiversity, so this work presents an important contribution to our survey of life in this region. The authors use existing ecological frameworks (e.g. isolation by distance, geographical parthenogenesis) to test specific hypotheses about the drivers of biodiversity of nematodes. The use of modelling approaches could be useful to help identify areas of this region that would be good to focus on for future soil biodiversity surveys.

The title is fairly broad, I would suggest including nematodes in the title to better convey the focus of this study.

A stated goal in the abstract was to study resilience in a unique ecosystem (desert), but I didn't see how "resiliency" was measured or assessed in this study.

Is the relationship between diversity and precipitation driven more by the fact that you had clusters of samples from 6 different sites (which also had different soil/mineral types, vegetation etc.)? How much of these other predictor variables were accounted for in the model? Is it possible to conclude that MAP is the driver, or is it simply due to spatial distances? Some focused within-transect modeling/testing may help strengthen and support the conclusion being drawn.

Similarly, with the relationship to latitude: I'm not an expert on this region, but the distribution of sample sites shows that the northern sites have more inland representation while the southern sites are closer to the coast – could that be affecting patterns by latitude?

How temporally stable are these patterns of diversity? This datasets represents a single time point; how much variation would you expect if you returned to those sites to sample?

How many samples were taken from each site? How much local spatial variability is expected at each location?

In general, the description of how the data was handled with respect to within- and between-transect analyses needs some more detail to clarify what was done (see specific comments for the methods section)

Specific suggestions by line number:

30 (abstract) - "desert" - lowercase; "indicators" instead of "indicatives"?

107-108 – capitalize taxa names

159 – not entirely clear what is meant by "we calculated the latter".

159-167 – it wasn't clear to me if the genetic distance results were within a geographical region or between regions. Unclear what "randomly" means here.

168 – subtitle of this section indicates random (no driving pattern), but then the paragraph goes on to suggest there is a relationship with geographic distance.

178 – no comma after "interested"

324 – "taxa are at an..."

349 – How many soil samples were taken at each site?

393 – Need more detail here on how the data was analyzed. How were the 6 transects treated in this analysis? Were the data from each site along the transect combined to get an aggregate community for the transect? Or means of each taxa calculated?

406 – Same comment as previous - what does "each transect" mean – a combination of the data from each site along a single transect? Mean? Other?

411 – which edaphic properties were included?

Fig S3 legend seems to be missing a reference

(Remarks on code availability)

README file is blank

Reviewer #3

(Remarks to the Author)

The authors present findings on ecological extremes and the impact on biodiversity, using nematodes as bioindicators.

The authors must be commended as this is a nicely written paper on a difficult but important theme and, from an extremely difficult terrain, logistically.

The results are original and solid for publication. There are limits to extrapolation from a relatively small sample size but the authors have not overdone any conclusions.

The methodology is sound however the M&M section would benefit with added detail on i) L351 how were the initially assessments (for nematodes on-site) done; ii) how many samples were taken at each of the six locations; iii) whether any systematic process (or not) was used to select individual sampling locations (bias?); and iv) it would be great to have a table with all 194 sampling GPS coordinates (if available).

One bit of advice is to re-read and be sure that it is nice and clear that this is a paper on the relationships between habitats and biodiversity, not a nematode paper. At times it strays a little too much about the nematode. But this is not a blocker.

There are a few typos so a couple of re-reads are required (e.g. L186, 236 and 248)

(Remarks on code availability)

Version 1:

Reviewer comments:

Reviewer #1

(Remarks to the Author)

I will not repeat my remarks from an earlier review. The authors have mainly addressed any suggestions or questions that I had.

I did request that the authors provide a full list of nematode taxa found. In the response, they say this information is in STable10, but it is not. That table only lists taxa that were subjected to 18S analysis. The paper reports that 21 families (36 genera) were found. This is the list I was expecting to see.

The paper still needs to be carefully read for comma placement and punctuation. Things that should be capitalized are not (eg, Firmicutes). Capitalization in the references needs attention.

(Remarks on code availability)

Reviewer #2

(Remarks to the Author)

The authors have done a thorough job of addressing reviewer comments in the revised manuscript. In particular, the additional modelling efforts to address within site variability and the attention given to other co-variates (e.g. vegetation, soils) in the discussion has strengthened the manuscript.

(Remarks on code availability)

Reviewer #3

(Remarks to the Author)

(Remarks on code availability)

Reviewer #4

(Remarks to the Author)

This study investigated soil nematode diversity across multiple levels - genetic, taxonomic, community, and life-cycle characteristics - in various Atacama Desert habitats including dune systems, high-altitude mountains, saline lakes, river valleys, and fog oases. The results that genus-level richness of soil nematodes exhibited a distinct latitudinal diversity gradient, showing positive correlation with increasing precipitation levels. The annual precipitation and temperature heterogeneity are the major drivers of genera richness. These findings are not surprising. The most interesting results is that asexual taxa is more likely to occur at higher altitudes, suggesting that elevation serves as the strongest predictor of reproductive strategy.

My major concerns:

1. There are too much supplementary tables and figures. It seemed that there were lots of data. However, the present tabular data and graphical representations prove challenging to decipher. The abundances of total nematodes, different trophic groups, family/genera/species, and the biodiversity of each sample are not enumerated, and the relationships between nematode variables and environmental variables are also unclear. Figure 4 and figure 1 are ambiguous. Figures of regression analysis between nematode and environmental variables are better than such figures.

2. How the nematode species was determined as asexual taxa or parthenogenesis?

Detailed comments:

L94-103: There is no need to describe the finding here.

Figure4: I am unfamiliar with this analysis. Which model makes the most accurate predictions?

L415-416: Two paragraphs?

L435: Was the top 0-10cm soil removed? Only the 10-30cm soil was collected?

Fig. S2: Panel D is missing.

(Remarks on code availability)

REVIEWER COMMENTS

1. Reviewer #1 (Remarks to the Author)

The authors conducted a survey of nematode communities in 231 soils from six different habitats from the Atacama Desert in Chile. They employed several techniques to gain ecological insights from this survey. First, they characterized the diversity (morphospecies/family/genus) of the communities through morphological analyses, augmented by partial 18S barcoding (haplotypes). Second, they modelled the structure of these communities incorporating several variables, including latitude, elevation, precipitation and temperature. Finally, they studied the reproductive habits of nematode that they were able to culture and correlated that to environmental characteristics.

Key results:

- 1. Characterization of the nematode taxa present in Atacama Desert soil – this is new information as no work has been done on these nematodes that I know of for over 50 years.*
- 2. Assessment of B-diversity (Jaccard Index) through incorporation of different sites with different characteristics (latitude, elevation, MAP)*
- 3. Finding a relationship between geographic and genetic distance for the genus *Panagrolaimus* through evaluation of biogeography and 18S genetic diversity of select taxa.*
- 4. Identification of positive effect of a latitudinal diversity gradient and increasing precipitation.*
- 5. A preliminary finding that asexual reproduction is positively correlated to elevation.*

Validity:

In so far as key results 1 & 2, the expertise of the team and the combination of morphological and barcoding approaches to identify the nematodes means the data is valid.

I think that use of the Bongers Index to characterize “poor soil food webs” that are “sensitive to disturbance” is a big stretch. The Index was developed mainly for agricultural systems and to investigate the impact of disturbance in those systems. I think, in the context of this study, the Index has little utility. It’s trying to impose what is known about the ecology of nematodes from eutrophic systems to the oligotrophic desert, and I don’t believe it is valid or necessary. Are any of these sites being actively disturbed in some manner? If not, saying the nematode communities are “sensitive to disturbance” isn’t really valid. While lower cp nematodes might be over-represented in a

disturbed agricultural soil, their presence in a desert soils is due to different phenomena.

Reply: We thank the reviewer for raising an important point in regard to “sensitive areas” in the Atacama Desert, as these are not imminently obvious to the reader. However, there are a multitude of threats to the ecosystem(s) there, such as for example the activity of mining or farming. For sites in our study, these threats were present in particular where more “sensitive” taxa were found. Consequently, we now mention mining, plant extraction, and uncontrolled tourism in the text for a better explanation (lines 311-316).

Furthermore, we are aware that the *Bongers index* was initially developed in an agricultural context. Our main goal in using the index is to provide a base line for further studies as this can impact conservation efforts in the area. We believe the index is indeed meaningful in this, as several publications (e.g. <https://doi.org/10.3897/mbmq.8.111307>, <https://doi.org/10.3389/fevo.2016.00084>, <https://doi.org/10.1002/ece3.8061>) have shown its utility in natural systems. The advantage of using an established index is that the interpretation of the value ranges is known, making it possible to understand the findings in the context of better-studied systems. Nevertheless, we agree that the oligotrophic conditions of the desert environment can have an impact on the index and for this we refer to a rather relative interpretation of our findings between sites within our study (lines 309-317).

Regarding key result 2, I think the modelling effort is interesting, but I found that some of the environmental data used was strange (see below). Discussion of how the soil habitat changes with the variables that were important was absent (eg, how do MAP, latitude, and elevation actually influence soils as a habitat for nematodes).

Reply: We thank the reviewer for pointing this out. We have now included more information about the effect of the main drivers on soil communities to complement the previous text. In the case of genera richness (taxonomic diversity per site), we highlight that precipitation is used as a proxy for water availability as done by other researchers previously, and how higher precipitation improves nutrient availability in soils, promotes higher bacterial densities (food source for nematodes) and increases soil moisture, which is necessary for nematodes to move as they live in the water film between soil particles (lines 335-338, 342-347). We also mention the effect of temperature variation (lines 348-354) and explain the theory of latitudinal diversity gradient, where underlying mechanisms for the pattern remain unclear (lines 355-369). Particularly for elevation,

Regarding key result 3: The sequencing effort seems valid and sufficient to be able to examine the haplotype diversity.

Reply: We want to mention here that some of the sequences in the analysis have now been removed due to insufficient quality and resolution for uploading them to Genbank. This reduced the number from 393 to 386. Hence, we repeated the calculations, but the conclusions remain the same.

Regarding 4: I think this is very, very interesting data, and I appreciate that the authors were candid in the paper regarding the limited sample size. I think saying in the abstract that the distribution of “asexual taxa appears to be in the marginal ranges of high altitudes” is not valid. As far as I can tell, you found a positive correlation with elevation for the small number of cultures established. The data does not show that asexual taxa

are not found at lower elevations.

Reply: We agree that the statement was not a true representation of the results and have corrected it in the text. This section now states, “We also find that distribution of asexual taxa is more likely to occur at higher altitudes” (line 30 -abstract), and we highlight in the discussion that there is a higher probability of asexual taxa to occur at higher elevations, this however does not imply that asexual taxa can’t be found at lower elevations. This can also be corroborated on 4 (D-E) where the probability of being asexual is closer to 1 (light orange to beige) with increasing elevation.

Significance:

Very few studies have attempted to examine nematode genetic diversity within taxa across sites, as the authors have done here. As well, very little work has been done to connect nematode reproductive strategy to the environment. So, these are good questions, and this is novel work of significance to the soil nematode community.

Data and methodology:

I think everything seems clear and appropriate.

Analytical approach:

Regarding the modelling effort to investigate environmental variables and nematode diversity: I think some edaphic and vegetation-related variables would have been useful in this effort, and I wondered if some of this data was collected. For example, soil properties (organic C is very important to nematodes) or NDVI. “Plant association” was mentioned at Line 228 as a variable, vegetation coverage and other variables are in Table S11. For me, the inclusion of Rock type and Soil thickness were not justified beyond being publicly available data (and indeed, these variables did not prove important in any of the models – I would have not put them in the paper at all – maybe leave them out of the figure, as done with topographic complexity and curvature, Line 416). To me, I think the significance of latitude, MAP and elevation on nematode communities is likely due to impacts on vegetation and thus soil organic matter (overall soil habitability), and I think these mechanisms should be at least discussed in the paper. I appreciate not wanting to include codependent variables in the modeling effort, however (but aren’t latitude and precipitation codependent? Surprising not, per S9).

Reply: We agree that plants can play an important role in the determination of nematode’s communities, and this is why we tried to include this information at a finer scale in the reproductive mode models from on-site measurements. For these models, we recorded the major plant groups present in the sampling localities, percentage of vegetation cover and type of coverage (shrubs, cacti and herbs). However, for the genera richness determination, we couldn’t define such fine scales (overall for the environmental variables tested) given that the reanalysis data implemented doesn’t not provide resolution at the sampling spot but rather on a grid. We discussed implementing NDVI information, but unfortunately, this satellite-based measures seem to have a large discrepancy between the real vegetation and the recorded vegetation, for instance, shrubs which are a major component of the areas we analysed are not recorded with NDVI. We thus rely on our more precise records, based on repeated extensive sampling efforts.

With regard to the choice of environmental variables, we have included more information in the discussion, why we selected rock type and soil thickness as variables that could influence nematode diversity (lines 355-359). We have also included a statement acknowledging that plant presence is an important factor for nematode diversity but clarify that we haven't looked into this for the genera richness aspect (lines 359-363). In short, we do consider plants to be an important factor in determining community diversity for nematodes, but have not included this in the modeling approach for the above mentioned reasons.

While latitude and precipitation show a weak positive correlation (0.43), we didn't consider the exclusion of these variables since the correlation was below $|0.65|$. For instance, mean annual temperature and elevation had a negative correlation of 0.79, hence only elevation was considered. This is now clarified in the methods (lines 517-519).

Suggested improvements:

What is a "transect" in this study? It seems that samples were collected haphazardly within sites, but "transect" implies some scheme that was not described in the paper. I would perhaps not describe the sampling as along a transect.

Reply: We thank the reviewer for pointing this out. It is true that we selected environmentally distinct areas, identified based on ecological and landscape features (e.g. dominant vegetation, presence of water bodies, different and landforms), rather than following strict transects (apart from Eagle Point, which is a transect on a mountain slope). We have been using the term transect as our sampling localities align with defined transects that have been established for the collaborative research cluster our project is part of (doi.org/10.1016/j.gloplacha.2020.103275). We have now changed the text to use the term "localities" and we specify criteria for selecting these as mentioned above.

Assuming the number was different for each site, can you provide a range for the number of samples taken? (Line 345) Also, can you clarify the degree of distance from plants that the samples were collected? Were you under the canopy? Was their litter?

Reply: We have included more details about sample collection in the Methods section (lines 412-425), i.e. samples taken near to plants corresponded to sediment approximately 10 cm away from the plant stem, or from plant litter in the case of Paposo. Plant litter are drying and dried leaves, twigs, potentially dried fruit found under a plant (explained in lines 434-435 in the text now). We state the number of samples per locality: 7 samples from Altiplano, 11 samples from Salars, 15 samples from Paposo, 20 samples from Totoral Dunes, 23 samples from Aroma and 21 samples from the Eagle point (lines 428-429) and provide sampling coordinates of all sampling spots (Supplementary tables S1 - S7)

I think that addition of a table in the paper or supplement that is a full list of the nematode family/genera and number of haplotypes found at each site is essential and would increase the utility of this work.

Reply: This information is now included in the supplementary files as table S10. The table, sorted by locality, contains the number haplotypes per taxonomic group. One

sequence (accession number PQ587770.1) was initially identified as Acrobeloides but after curation corrected to Acrobeles Hap_12. Due to uncertainties in the differentiation between Aporcelaimidae, Qudsianematidae and Pararhysocolpidae based on the current database only the order level was assigned for most identifications.

I would leave off the Bongers analysis – I don't think it is informative.

Reply: See our comments above for a justification of the use of this index. Nevertheless, we have reduced the emphasis on these results.

Clarity and context:

I have made some comments above as to where I think more context would be helpful (interpretation of modelling results).

In Figure A: What is “temperature range”? Is that yearly? This is just not a variable I am familiar with.

Reply: We have included an explanation of how the temperature range was defined: as the difference between seasonal (rather than daily) minimum and maximum values, serving as a proxy for climatic heterogeneity.

Lines 153-154: I have read this many times and I cannot make sense of this text, but I feel like maybe it's important.

Reply: This statement has been changed to make it clearer “Overall, we were not able to find any distinct geographical clustering of haplotypes based on the networks, i.e. individual haplotypes were present in geographically distant localities” (lines 154-155).

Figure 3: Remove UEF from legend – presence is confusing and unnecessary.

Reply: UEF has been removed from the legend.

2. Reviewer #2 (Remarks to the Author):

This manuscript presents a comprehensive investigation of nematode diversity in various subregions of the Atacama. Diversity (genetic, taxonomic, reproductive strategies, functional groups) are investigated and statistical approaches used to determine best predictors (elevation, MAP) of those patterns. This is a unique environment, with limited investigations of the soil biodiversity, so this work presents an important contribution to our survey of life in this region. The authors use existing ecological frameworks (e.g. isolation by distance, geographical parthenogenesis) to test specific hypotheses about the drivers of biodiversity of nematodes. The use of modelling approaches could be useful to help identify areas of this region that would be good to focus on for future soil biodiversity surveys. The title is fairly broad, I would suggest including nematodes in the title to better convey the focus of this study.

Reply: We have included the term “Nematode” in the title for clarity. However, we would like to point to the comments by reviewer 3 here, who asked for less focus on nematodes in the text in general. We have now aimed to balance this better, acknowledging that our study is based on nematodes, but inferences can potentially be extended to soil invertebrates.

A stated goal in the abstract was to study resilience in a unique ecosystem (desert), but I didn't see how “resiliency” was measured or assessed in this study. Is the relationship between diversity and precipitation driven more by the fact that you had clusters of samples from 6 different sites (which also had different soil/mineral types, vegetation etc.)? How much of these other predictor variables were accounted for in the model? Is it possible to conclude that MAP is the driver, or is it simply due to spatial distances? Some focused within-transect modeling/testing may help strengthen and support the conclusion being drawn.

Reply: We previously hadn't consider within transect testing, and we thank the reviewers for pointing it out as an important concern, as there could be clustering from a sampling bias. In order to address this, we went back to our data and re-grouped the samples per locality (formerly named transects) resulting in a total of 112 samples.

Specifically, we combined some subsamples of the Eagle Point Transect as these were taken in very close proximity with no distinctive ecological features (e.g. no proximity to different plant species, rock or different soil type). This ensures better comparability of this sampling point with all other samples in the analysis.

We then tested our dataset, as previously done in the first version of the manuscript, using a test of normality, PCAs, and variable correlation, before proceeding test the GLS and Random Forest models again. This test confirmed our previous conclusion that according to the random forest model the most important variables in determining genera richness are elevation, latitude and mean annual precipitation (with the addition of temperature range). We can accordingly exclude a clustering bias in our data.

As the GLS model seems to overestimate genera richness, not accounting for sampling bias and locality selection, we tested 57 more models using a linear mixed-effects model approach. Overall, these confirmed that climate is a predictor for genera richness in the desert given. See lines 524-527 (Methods), lines 219-229 (Results), and lines 335-354 (Discussion).

Regarding spatial distances and their potential role as the primary driver of diversity instead of mean annual precipitation, we found a moderate isolation-by-distance pattern when analyzing community dissimilarity (Mantel test: $\rho=0.5153$, $p=0.048611$, 719 permutations) (lines 176–178). However, this pattern depends on the specific taxa inhabiting each area. Since our model focuses on the number of different taxa rather than specific genera or families, we believed this didn't directly impact the results. We have now performed a correlation test specifically between geographic distance of the localities and the average precipitation per locality, the result showed that there is no significant correlation between these two characteristics ($\rho=0.1321$, $p=0.24861$, 719 permutations), discarding that geographical distance is correlated to precipitation, geographical distance does not reflect the precipitation patterns. Moreover, thanks to reviewer comments with included a modeling approach accounting for sampling locality as depicted above.

We also tested the effect of the locality in the reproductive mode models, but this had no significant effect.

Similarly, with the relationship to latitude: I'm not an expert on this region, but the distribution of sample sites shows that the northern sites have more inland representation while the southern sites are closer to the coast – could that be affecting patterns by latitude? How temporally stable are these patterns of diversity? This datasets represents a single time point; how much variation would you expect if you returned to those sites to sample? How many samples were taken from each site? How much local spatial variability is expected at each location? In general, the description of how the data was handled with respect to within- and between-transect analyses needs some more detail to clarify what was done (see specific comments for the methods section)

REPLY: We have performed the analysis including *locality* as a random effect as detailed above. Furthermore, it is important to consider that in Chile the distribution from the coast to the more in-land locations are well-described by the altitudinal variation. The coastal cordillera is marked by a large altitudinal increase closer to the coast and the altiplano then marks the plateau area at higher elevation before the uprise of the Andes. That way, the longitudinal distribution of localities is reflected by variation in altitude.

We have clarified how stable we expect this patterns to be: on shorter time-scales climatic variation is present (seasonal patterns), our results reveal nematode diversity across different levels but do not provide insights into abundance, which can be expected to be highly variable across seasons (e.g., during the Altiplano rainfall season). Nevertheless, we expect that taxonomic diversity remains relatively stable throughout the year for the localities and sampling points analyzed, given that several taxa are capable of undergoing anhydrobiosis (lines 400-405).

For better clarity, the number of samples per locality is now specified in the materials and methods (lines 428-431).

Specific suggestions by line number:

30 (abstract) - "desert" - lowercase

Reply: Has been changed

“indicators” instead of “indicatives”?

Reply: Has been changed

107-108 – capitalize taxa names

Reply: Has been changed: Proteobacteria and Actinobacteria.

159 – not entirely clear what is meant by “we calculated the latter”.

Reply: We changed it to Euclidian distance for clarity (lines 160-161).

159-167 – it wasn’t clear to me if the genetic distance results were within a geographical region or between regions. Unclear what “randomly” means here.

168 – subtitle of this section indicates random (no driving pattern), but then the paragraph goes on to suggest there is a relationship with geographic distance.

Reply: We thank the reviewer for pointing this out, we have rephrased the title for a better representation of moderate effect of geographical distance.

178 – no comma after “interested”³²⁴ – “taxa are at an...”

Reply: Has been corrected in the text.

349 – How many soil samples were taken at each site?

Reply: We have included the number of samples taken at each spot in the materials and methods (lines 428-429).

393 – Need more detail here on how the data was analyzed. How were the 6 transects treated in this analysis? Were the data from each site along the transect combined to get an aggregate community for the transect? Or means of each taxa calculated?

Reply: We acknowledge this wasn’t clear in the methods, we have included a statement clarifying that we analyzed the communities in the localities (see above for a description on why we use the term “locality” now) as aggregates since we only relied on presence absence data but not abundances. This information can be found on lines 480-482.

406 – Same comment as previous - what does “each transect” mean – a combination of the data from each site along a single transect? Mean? Other?

Reply: We have now specified how we selected and defined the different sampling localities. The rationale is explained in a reply to ref 1 above and in the text in lines 421-425.

411 – which edaphic properties were included? Fig S3 legend seems to be missing a reference

Reply: We analysed the soil thickness (depth to mother rock) as edaphic property and the type of mother rock (parental material). The reference in the supplementary figure has been included.

Reviewer #2 (Remarks on code availability):

README file is blank

Reply: We apologize for this faulty file. The readme file has been updated and now includes all information where each of the analysis can be found.

3. Reviewer #3 (Remarks to the Author):

The authors present findings on ecological extremes and the impact on biodiversity, using nematodes as bioindicators.

The authors must be commended as this is a nicely written paper on a difficult but important theme and, from an extremely difficult terrain, logistically.

The results are original and solid for publication. There are limits to extrapolation from a relatively small sample size but the authors have not overdone any conclusions.

The methodology is sound however the M&M section would benefit with added detail on
i) L351 how were the initially assessments (for nematodes on-site) done

Reply The procedure of on-site identification of presence-absence of nematodes was the same as for the laboratory checking for further morphological and molecular work (we brought the Zeiss Stemi 2000 to the field). This is now also clarified in the text (line 435).

ii) ii) how many samples were taken at each of the six locations

Reply: We have included the number of samples taken at each spot in the materials and methods (lines 428-429)

iii) whether any systematic process (or not) was used to select individual sampling locations (bias?)

Reply: We have now specified how we (systematically) selected and defined the different sampling localities. The rationale is explained in a reply to reviewer 1 above and in the text in lines 421-428. In general sampling was conducted in a systematic way by sampling in straight lines with evenly spaced out distances in at each of the localities.

iv) it would be great to have a table with all 194 sampling GPS coordinates (if available).

Reply: We have included 6 supplementary tables displaying the exact coordinates of each sample analysed from the different localities as well as one table with coordinates from samples outside of those defined areas (tables S1-S6 of supplementary material).

Reply: As we decided to group samples in localities and test for sampling bias using extra 57 models with a linear mixed-effects approach that account for the effect of the sampling locations, we noticed that subsets of samples from EPT should be grouped together: e.g. EPTA01, EPTA02, EPTA03 have been grouped as sample EPTA to account for 500g of sediment in specific coordinate set that shares all identical environmental measures. These samples were analyzed in subsets, hence we initially had them labelled differently, but noticed upon revision that this wouldn't guarantee full comparability with all other samples. In the analysis they are now grouped as explained above (grouped

by close proximity where no distinctive features, e.g. plant nearby, were present). Therefore, the number of total samples is 112 as can be seen in the tables with coordinates. All analysis has been repeated where this could have an impact, and it has been specified in lines 524-526 (Methods), lines 219-229 (Results), and lines 342-354 (Discussion).

One bit of advice is to re-read and be sure that it is nice and clear that this is a paper on the relationships between habitats and biodiversity, not a nematode paper. At times it strays a little too much about the nematode. But this is not a blocker.

Reply: Thank you for the important suggestion, we appreciate it, as it is far too easy to drift off into describing these fascinating animals. We have gone through the text to shift the focus even more towards habitats and biodiversity. However, to make clear that we used nematodes as our study system, and based on the suggestion of reviewer 2, we added the term nematode to the title.

There are a few typos so a couple of re-reads are required (e.g. L186, 236 and 248)

Reply: We combed through the text to fix typos. Thank you for pointing these out.

Point-by-point response to reviewers

Reviewer #1 (Remarks to the Author):

I will not repeat my remarks from an earlier review. The authors have mainly addressed any suggestions or questions that I had.

I did request that the authors provide a full list of nematode taxa found. In the response, they say this information is in STable10, but it is not. That table only lists taxa that were subjected to 18S analysis. The paper reports that 21 families (36 genera) were found. This is the list I was expecting to see.

The paper still needs to be carefully read for comma placement and punctuation. Things that should be capitalized are not (eg, Firmicutes). Capitalization in the references needs attention.

We thank the reviewer for their thoughtful comments and for highlighting the lack of clear access to the requested dataset. While the data were previously included as part of the model input in Code Ocean (10.24433/CO.6395549.v2), we acknowledge that this was not an intuitive or easily accessible location. To improve transparency and accessibility, we have now deposited the dataset in a Zenodo repository (<https://doi.org/10.5281/zenodo.14930774>), under the file name "**FamilyGeneraPerSample**". This table contains sample coordinates, sample IDs, and the identified families and genera. We have also revised the main text to clearly reference this dataset (line135).

Reviewer #2 (Remarks to the Author):

The authors have done a thorough job of addressing reviewer comments in the revised manuscript. In particular, the additional modelling efforts to address within site variability and the attention given to other co-variates (e.g. vegetation, soils) in the discussion has strengthened the manuscript.

We would like to thank again the reviewer #2, we also feel implementing his previous comments improved the analysis.

Reviewer #4 (Remarks to the Author):

This study investigated soil nematode diversity across multiple levels - genetic, taxonomic, community, and life-cycle characteristics - in various Atacama Desert habitats including dune systems, high-altitude mountains, saline lakes, river valleys, and fog oases. The results that genus-level richness of soil nematodes exhibited a distinct latitudinal diversity gradient, showing positive correlation with increasing precipitation levels. The annual precipitation and temperature heterogeneity are the major drivers of genera richness. These findings are not surprising. The most interesting results is that asexual taxa is more likely to occur at higher altitudes, suggesting that elevation serves as the strongest predictor of reproductive strategy.

My major concerns:

- 1. There are too much supplementary tables and figures. It seemed that there were lots of data. However, the present tabular data and graphical representations prove challenging to decipher. The abundances of total nematodes, different trophic groups, family/genera/species, and the biodiversity of each sample are not enumerated, and the relationships between nematode variables and environmental variables are also unclear. Figure 4 and figure 1 are ambiguous. Figures of regression analysis between nematode and environmental variables are better than such figures.***

We thank the reviewer for highlighting areas where the presentation of results could be improved for clarity. As noted in our response to Reviewer #1, a table listing all identified families and genera for each sampling location is now explicitly referenced in the main text (line 135) and is available via Zenodo (<https://doi.org/10.5281/zenodo.14930774>). Regarding nematode abundance, our study does not provide quantitative insights due to the temporal variability in sampling (i.e., different seasons), which could introduce inconsistencies in abundance data. Instead, we focused on identifying the adult taxa present in each sample. As discussed in the manuscript (lines 409–414), while we expect taxonomic composition to remain relatively stable throughout the year, nematode abundance is likely to fluctuate substantially with seasonal dynamics, particularly during events such as the Altiplano rainfall season.

To improve the clarity of Figures 1 and 4, we have revised their legends to better articulate their purposes: Figure 1 contextualizes the climatic and geological (geomorphological) characteristics of the Atacama Desert, while Figure 4 illustrates how predictions vary spatially depending on the modeling approach and variables used. Additionally, to address the reviewer's concern about data presentation, we have added the figures of the regression analysis between nematode diversity and environmental variables in the supplementary material (Supplementary Figure S11). These plots highlight the generally weak linear associations when considering individual predictors, which is the main reason we used modelling approaches less sensitive to these non-linear effects, such as random forest. To complement this, we include partial dependence plots (Supplementary Figure

S12), which illustrate the non-linear effects of key predictors within the random forest model framework.

2. *How the nematode species was determined as asexual taxa or parthenogenesis?*

Nematodes reproductive mode was assessed following the protocol from Lewis 209 and this is specified in the text in lines 555-563.

Detailed comments:

3. *L94-103: There is no need to describe the finding here.*

We appreciate the reviewer's suggestion. However, we would prefer to retain this section in the introduction, as we believe it helps set the stage for the reader by providing context and a clear sense of the study's direction and relevance.

4. *Figure4: I am unfamiliar with this analysis. Which model makes the most accurate predictions?*

We have included a statement acknowledging that, due to the differing assumptions underlying each modeling approach and the limited set of observations used, it is not possible to select a single "best" model at the moment. Therefore, we focus on identifying consistent trends across modelling approaches. Notably, genus richness shows responses to environmental gradients that are not fully captured by linear models alone. Thus, at this stage, random forest models could be considered as the most accurate, due to the nonlinear effects observed in the observations. However, temperature heterogeneity consistently emerged as a strong predictor across all modeling approaches. Additionally, mean annual precipitation was a significant factor in both linear mixed-effects and random forest models, highlighting the key role of climate in shaping genus richness (lines 233–238). These preferred models are contingent on our set of observations, as new information about nematode diversity emerges from the region, these models should be revised with the up-to-date observations.

L415-416: Two paragraphs?

5. *L435: Was the top 0-10cm soil removed? Only the 10-30cm soil was collected?*

We now clarify in the text (line 443) that samples were collected from depths ranging between the upper 10 cm (0–10 cm) and up to 30 cm, depending on soil characteristics. The final depth was determined by factors such as soil coarseness and the point at which rock was encountered, limiting further sampling.

6. Fig. S2: Panel D is missing.

The figure has been fixed and the letter labelling is correct now.